# Acute Effects of Fitlight Training on Cognitive-Motor Processes in Young Basketball Players

**DOI:** 10.3390/ijerph20010817

**Published:** 2023-01-01

**Authors:** Fioretta Silvestri, Matteo Campanella, Maurizio Bertollo, Maicon Rodrigues Albuquerque, Valerio Bonavolontà, Fabrizio Perroni, Carlo Baldari, Laura Guidetti, Davide Curzi

**Affiliations:** 1Department Unicusano, University “Niccolò Cusano”, 00166 Rome, Italy; 2Department of Theoretical and Applied Sciences, eCampus University, 22060 Novedrate, Italy; 3BIND-Behavioral Imaging and Neural Dynamics Center, Department of Medicine and Aging Sciences, University G. d’Annunzio of Chieti-Pescara, 66100 Chieti, Italy; 4Neurosciences of Physical Activity and Sports Research Group, Department of Sports, Universidade Federal de Minas Gerais, Belo Horizonte 31120-901, Brazil; 5Department of Biotechnological and Applied Clinical Science, University of L’Aquila, 67100 L’Aquila, Italy; 6Department of Biomolecular Sciences, University of Urbino Carlo Bo, 61029 Urbino, Italy

**Keywords:** executive function, cognitive-motor training, massed training, exercise, fitness, perceived effort

## Abstract

Cognitive-motor training could be used to improve open-skill sport performances, increasing cognitive demands to stimulate executive function (EF) development. Nevertheless, a distributed training proposal for the improvement of EFs is increasingly difficult to combine with seasonal sport commitments. This study aimed to investigate whether a massed basketball training program enriched with Fitlight training can improve EFs and motor performance. Forty-nine players (age = 15.0 ± 1.5 yrs) were assigned to the control and Fitlight-trained (FITL) groups, which performed 3 weeks of massed basketball practice, including 25 min per day of shooting sessions or Fitlight training, respectively. All athletes were tested in cognitive tasks (Flanker/Reverse Flanker; Digit Span) and fitness tests (Agility T-test; Yo-Yo IR1). During the intervention, exercise/session perceived effort (eRPE/sRPE) and enjoyment were collected. RM-ANOVA showed significant EFs scores increased in both groups over time, without differences between the groups. Moreover, an increased sRPE and eRPE appeared in the FITL group (*p* = 0.0001; *p* = 0.01), with no group differences in activity enjoyment and fitness tests. Three weeks of massed basketball training improved EFs and motor performance in young players. The additional Fitlight training increased the perceived cognitive effort without decreasing enjoyment, even if it seems unable to induce additional improvements in EFs.

## 1. Introduction

Executive functions (EFs) are a family of cognitive functions that are essential in a high number of daily life activities, and they include inhibition (including selective attention), working memory and cognitive flexibility (mental shifting and creativity). These functions, called core EFs, are essential for developing high-order EFs, such as problem solving, reasoning and planning [1].

From a neurophysiological point of view, these functions share the same cerebral area with some motor circuits such as motor planning or performance of complex motor tasks: the prefrontal cortex [2]. Researchers have suggested that physical activity is one of those activities that, with the correct stimuli, could improve EFs [3,4]. In fact, the development of EFs, especially in children and adolescents, may be associated with sports practice, since sport activities usually require not only physical but also cognitive involvement [5,6]. Recent literature shows that the development of EFs is age-related and increases according to an athlete’s growth [7]. However, the influence of age on specific EF parameters is still debated (e. g. accuracy of the response) [7,8].

In a recent meta-analysis, Contreras-Osorio and colleagues [6] analyzed different sports programs that could possibly affect executive function in students. The authors found that team sports, enriched with tasks that stimulate cognitive development, could increase the involvement of EFs compared to individual sports [6]. In agreement with them, Waelle and colleagues [9] concluded that children involved in team sports show superior levels of EF development compared to those practicing other self-paced sports. Regarding the different contribution of open- and closed-skill exercises on EF development, a recent systematic review reported that open-skill exercises showed superior effects compared with closed-skill exercises on EF enhancement in both children and adults [10].

During open-skill sports practice, a huge number of variables and stimuli should be considered, which can increase the cognitive involvement of EFs (such as working memory or inhibition of stimuli) [5,6]. In fact, the complexity of the open-skill sport environment plays a key role in the development of both the motor and cognitive areas [11,12].

It has also been suggested that, if on the one hand open-skill sports practice is able to induce EF development, on the other, having a superior level of EFs has also been correlated with an athlete’s better performance in many open-skill sports, such as volleyball, table tennis, soccer, tennis and basketball [13,14,15,16,17,18].

However, Heilmann et al. [19] in their recent meta-analysis highlighted how the increased EF involvement seemed to be more related to a sport’s cognitive demands, than to the simple differentiation between open- or closed-skill exercises [19]. Basketball is an open-skill sport with high cognitive demands that stimulates and simultaneously requires EF activation during match actions. Indeed, basketball players continuously need to discriminate between relevant and irrelevant stimuli quickly (1–2 s) [20]. The capability to respond correctly requires critical focus, selective attention and inhibitory control of conflicting information [21].

Thus, the enhancement of EF levels in basketball athletes could lead to an improvement in decision-making during sport actions. In fact, decision-making is needed in match rapid action changes that occur in a limited time, characterizing open-skill sports [22]. This process is determined by the decision strategies and the EFs available, since making the right decision requires information search and processing [23]. On theoretical basis, a better working memory is needed to generate response options during the game, improving the quality of the first options, which are likely to be selected in action responses [24]. Moreover, having a high level of inhibitory control can lead to stopping option generation, thus leading athletes to focus on fewer, better-quality options [25]. Hence, athletes who are able to quickly generate high-quality options and inhibit low-quality responses, are likely to succeed in decision-making strategies. Furthermore, the capability to rapidly switch between response options and adapt to new situations is also determined by the level of cognitive flexibility [1]. On the basis of this theoretical knowledge, and in line with recent studies [22], we can assume that improving core EFs, could lead to improving decision-making strategies, and thus athletes’ performance in open-skill sports.

To enrich basketball practice with cognitive-motor training (CMT), sensorized light systems, such as Fitlight trainer, have been used in recent studies to examine their effects on EF development [26,27]. During physical training these kinds of devices are able to interact with users providing interactive and challenging tasks and improving enjoyment during task performances. Thanks to this feature a wide range of motor drills have been built to trigger cognitive function activation (such as attention, working-memory, inhibition and cognitive flexibility) [26]. Lucia and colleagues confirmed the efficacy of CMT training protocols on sports performance and cognition compared to training based on motor exercises only [26]. However, they extended the literature showing that these effects might be explained by enhanced anticipatory brain processing in the prefrontal cortex. Moreover, the Fitlight training system ^TM^ (2011) [28] has been proven to stimulate EF involvement during different sport activities optimizing human reaction times in team-game athletes. Specifically, Badau et al. [27] found that these athletes, following a 12-week program of exergame exercises using Fitlight technology (three times a week; 30 min per training session), showed shorter reaction times in computerized tests.

To assess the effects of a massed CMT program on EFs and motor performance, this study investigated whether the Fitlight training system ^TM^, used to cognitively enrich a massed basketball training program, can improve EFs (in particular, response inhibition, working memory and cognitive flexibility) and motor performance (i.e., agility and aerobic capacity) in young athletes. Moreover, the rate of perceived effort and enjoyment was evaluated during basketball practice to better understand athletes’ CMT perception using the Fitlight technology. The EF improvement during adolescence, a crucial period for both motor and cognitive development, could be a key factor to positively affect further improvement in open-skill sports performance.

## 2. Materials and Methods

### 2.1. Participants and Study Design

A randomized interventional study was performed. Fifty-eight male basketball players (mean ± SD, age: 15 ± 1.5 years; weight: 64.1 ± 13.6 kg; height: 173.8 ± 10 cm; sitting height: 88.1 ± 6.4 cm) were included in this study. Participants were recruited from three different Italian basketball clubs (U.S.D. Pallacanestro Urbania (n: 26), Metauro Basket Academy (n: 18) and C.S. 93 Basket Vadese (n: 5)) competing in the same youth categories, the Inter-Regional Basketball Championships U17-U15, organized by the Italian Basketball Federation (FIP). All participants voluntarily participated in the research. The puberal development was evaluated by a self-administered rating scale for a pubertal development questionnaire used in previous published studies [29,30]. The pubertal development stage predominant among the players was the mid-pubertal stage.

All subjects were randomly assigned to two experimental groups: control (CTRL) and Fitlight-trained (FITL) group. Regardless of the club membership, athletes belonging to the same experimental group were trained together for the whole intervention period in the same basketball stadium. At the end of the intervention, nine players were excluded from the study due to injuries, COVID-19 vaccinations and illnesses, that compromised their continued participation in the sport (four and five athletes from the U.S.D. Pallacanestro Urbania and the Metauro Basket Academy, respectively). Forty-nine athletes completed the training protocol: 24 players in the control (CTRL) and 25 in the Fitlight-trained (FITL) group. To be eligible, players were required to be healthy, without any physical or psychological difficulties that could affect the study.

Additional inclusion criteria were: (a) to be male; (b) to have been involved in competitive and training basketball activities for at least 5 years before the project, showing good temporal continuity (at least 75 min of basketball training, three times a week); (c) to be available daily to move for training in an equivalent way according to the experimental groups.

The following exclusion criteria were applied: (a) any condition, disease or therapy that could compromise the safety of volunteers while exercising; (b) taking medication, nutritional supplements or drugs; (c) smoking or drinking alcohol.

This study was reviewed and approved by the Institutional Review Board of the University of eCampus (registered number: 02/2021) in accordance with the ethical standards of the institutional and/or national research committee and with the 1964 Helsinki declaration and its later amendments or comparable ethical standards.

After being informed about the project, with the additional issue of a written information document, the participants and their parents provided informed and written consent before participation in the study.

### 2.2. Experimental Procedures

The research was carried out during the pre-season period. For the project, all participants had trained for three weeks, after two weeks of specific training to avoid injuries and to permit better adaptation to the imminent load increase. During the three weeks, both groups were trained for 5 days per week (from Monday to Friday) with two days of rest (Saturday and Sunday). The schedule always included a single training session per day (4:00–6:00 p.m.). To avoid any kind of different treatment between the experimental groups, the same training content and methodologies were used, except for the Fitlight training sessions which were exclusively employed in the FITL group, while the CTRL group performed couple or group shooting drill sessions.

In particular, Fitlight training consisted of footwork, shooting and dribbling drills, where the Fitlight training system was used to affect decision-making, hand–eye coordination and peripheral awareness (Table 1). On the other hand, shooting sessions included similar content but the main goals concerned were improvement in shooting technique and mental toughness.

Each session lasted 25 min and chronologically followed the warm-up phase at the beginning of each workout. All subjects submitted to the following tests according to the timeline of Figure 1.

### 2.3. Measurements

#### 2.3.1. Agility T-Test

The agility T-test [31,32] was used to evaluate agility in the players’ running performance. For the test, four cones were arranged in a T shape. Three cones were placed 5 m apart in a straight line. The starting cone was placed 10 m away, perpendicularly extending to the middle cone. Participants were asked to accelerate to touch each cone base and run forwards, laterally, and backwards between the cones as fast as possible. A dual infrared reflex photoelectric cells system (Polifemo, Microgate, Udine, Italy) was used to evaluate the performance time.

#### 2.3.2. Yo-Yo Intermittent Recovery 1 (Yo-Yo IR1) Test

Yo-Yo IR1 test [33] was used to measure the capacity to carry out intermittent exercise to stimulate maximal activation of the aerobic system. The test consisted of repeated 2 × 20 m sprints between a starting, turning, and finishing line at a progressively increasing speed determined by audio bleeps from an audio system. Between each sprint, subjects had a 10 s active recovery period, walking back and forth in a 2 m line marked by cones behind the start/finishing line. When a subject failed to cross the finish line before the bleep, a warning was given. When a subject failed for a second time to cross the finish line before the bleep, the test was considered concluded, and the final distance covered was registered and represented the end result.

#### 2.3.3. Physical Activity Enjoyment Scale (PACES)

Enjoyment was measured using the PACES questionnaire. This adapted version, developed by Motl et al. [34], consisted of 16 items (9 positive and 7 negative poled items) with responses on a 5-point Likert scale (1 = “I disagree a lot”; 5 = “I agree a lot”). All items regarded feelings about physical activity enjoyment suggesting face validity of the questionnaire. For the overall scale, negatively worded items were recorded to fit with the positively worded scale. Then, the average of the sum of items was calculated.

#### 2.3.4. Borg’s CR-10 Scale

Both exercise and session ratings of perceived exertion (eRPE and sRPE) were monitored by means of the Borg’s CR-10 scale [35]. While eRPE referred to the perceived exertion related to the Fitlight training or shooting sessions in the FITL or CTRL groups, respectively, sRPE referred to the whole training session in both experimental groups. To assess RPE during the exercise sessions, standard instructions and anchoring procedures were explained during the familiarization session. A rating of 0 was associated with no effort (rest) and a rating of 10 was considered to be maximal effort and associated with the most stressful exercise performed.

#### 2.3.5. Flanker/Reverse Flanker Task

The computerized version of the Flanker/Reverse Flanker task was used [36,37]. This test consisted of three different consecutive blocks, and a series of five fishes (blue or pink) was displayed in each one. The first one shows the classic Flanker paradigm (Eriksen and Eriksen 1974), where all the fishes were blue and the participants had to indicate the correct direction of the central stimulus, selectively responding and ignoring the flanking stimuli. Participants were asked to press the rightmost key if the central stimulus was pointing right and the leftmost key if the central stimulus was pointing left.

The second block presented a Reverse Flanker condition, the five fishes were pink, and the rule was to press the key corresponding to the fishes outside the central stimulus, ignoring the central stimulus.

For block 3 (mixed), there was a random alternation between the blue and pink fishes, keeping the rules for the fish’s color. Therefore, it seemed that the test required attentional control, inhibiting prepotent responses, re-orienting where to focus one’s attention, and remembering both rules [1,36,38]. So, the tests assessed the core EFs (working memory, inhibitory control and cognitive flexibility).

The participants performed 22 trials (16 congruent and 6 incongruent) in the first two blocks, while the third block was composed of 44 trials (32 congruent and 12 incongruent), corresponding to a total of 88 trials. Each block was preceded by some practice trials that were not computed into the analysis (flanker and reverse flanker: 4 trials; mixed block: 8 trials), where the volunteer received visual feedback.

For the analysis, only the third block was used. The percentage of correct responses (accuracy) and the average response time (RT) were analyzed. For these analyses, all trials in which RT was <250 ms were invalid because the players were unable to perceive the stimulus and adequately inhibit a response before the stimulus was processed. After excluding these trials, the percentage of correct responses on valid responses from block 3 was calculated. To calculate the mean RT from block 3, all trials in which the RT exceeded the upper or lower threshold of ±2 standard deviations were excluded.

More details about the task can be found by Hooper et al. [37].

#### 2.3.6. Digit Span Task

The Digit Span task used for this research consisted of two different tasks: the Forward-Digit Span and the Backward-Digit Span. The first measured short-term memory, while the second measured working memory [1].

The participants read a series of digits on a PC screen, at a rate of one digit per second, and were required to digit them on a PC keyboard following a specific order. If they wrote the correct order, they were given a longer list. The number of digits increased by one until the participant consecutively failed two trials of the same digit span length. The length of the longest list a person could remember represented the person’s digit span. In the Forward-Digit Span, participants were asked to repeat back the items in the order in which they read them. On the contrary, in the Backward-Digit Span, participants were required to write the digits in reverse order [39].

The span score corresponded to the maximum number of digits successfully reached. Additionally, we calculated the average response time (RT) and the rate correct score (RCS), defined as the span divided by the average RT [40].

### 2.4. Statistical Analysis

The distribution of each variable was examined using Shapiro–Wilk tests. Mean and standard deviation were used as a descriptive statistical approach. Two-way repeated measures ANOVAs were used. The sphericity assumption was tested using the Mauchly’s test, and the Greenhouse–Geisser correction was used when the sphericity assumption was violated. Additionally, when two-way repeated measures ANOVA assumptions were violated, the F1-LD-F1-model of the ANOVA-type statistics for non-parametric longitudinal data analysis from the nparLD R package were used. Bonferroni post hoc tests were used when appropriate. Partial eta-squared (ηp²) was used as a measure of effect size and classified using the following scale: small, ≥0.01 and <0.09; medium, ≥0.09 and <0.25; and large, ≥0.25. The significance level was set at α < 5%, and all analyses were performed in RStudio Version 1.1.463 for Windows, an integrated development environment for R.

## 3. Results

### 3.1. Executive Functions

Overall, both groups improved significantly in the EF tests over time. However, no significant differences were found in the EF tests between the FITL and CTRL, as well as in the interaction between the groups and time.

#### 3.1.1. Flanker/Reverse Flanker Task

The Flanker/Reverse Flanker task displayed a significant improvement of the assessed outcomes in both groups over time (from T0 to T8). The accuracy (Figure 2A) of the responses increased significantly over time in both groups [F(5.22, 246) = 7.50; *p* < 0.001; ηp² = 0.14]), with a medium effect size. No significant differences in the groups [F(1, 47) = 0.04; *p* = 0.842; ηp² < 0.01] and the interaction between the groups and time [F(5.22, 246) = 1.09; *p* = 0.369; ηp² = 0.02] were found. In addition, the response time (Figure 2B) decreased significantly over time in both groups [F(8,376) = 43.08; *p* < 0.001; ηp² = 0.48], with a large effect size. No significant differences in the groups [F(1, 47) = 0.07; *p* = 0.786; ηp² < 0.01] and the interaction between the groups and time [F(8, 376) = 1.12; *p* = 0.345; ηp² = 0.02] were found.

#### 3.1.2. Forward- and Backward-Digit Span 

In the working memory assessment by Forward- and Backward-Digit Span (Figure 3) over time, both groups significantly enhanced their span scores (Figure 3A,D) in forwards [F = 2.97; *p* = 0.004; ηp² = 0.05] and backwards [F = 4.60; *p* < 0.001; ηp² = 0.09].

In addition, both groups significantly improved their response times (Figure 3B,E) of the correctly recalled answers over time forwards [F(8, 376) = 3.70; *p* < 0.001; ηp² = 0.07] and backwards [F(5.9,278) = 2.70; *p* = 0.015; ηp² = 0.05]. No significant differences in the groups and the interaction between the groups and time were found in span or response time measures, forwards or backwards. Additionally, both groups significantly improved the rate of correct scores (Figure 3C,F) in the first time measures, forwards [F(6.1, 286) = 12.6; *p* < 0.001; ηp² = 0.21] and backwards [F(5.1,238) = 9.90; *p* < 0.001; ηp² = 0.17]. Moreover, the analysis of the rate of correct scores showed a significant interaction between the groups and time [F(6.1, 286) = 2.9; *p* = 0.008; ηp² = 0.06] in the Forward-Digit Span. The FITL group showed an increased rate of correct scores during the whole training sessions, higher than the CTRL group.

### 3.2. Perceived Effort

The analysis of the eRPE from T1 to T6, showed a significant difference in the interaction between the groups and time [F = 2.86; *p* = 4.69; ηp² = 0.03]. The FITL group revealed a significantly (*p* = 0.01) higher perceived effort in the group: time analysis, as shown in Figure 4A. No significant differences in the groups [F = 1.64; *p* = 0.200; ηp² = 0.02] and time [F = 1.00; p = 0.412; ηp² = 0.02] were found (Figure 4A). Moreover, the analysis of the sRPE (Figure 4B) from T1 to T6, showed a significant difference in time [F = 4.16; *p* = 0.002; ηp² = 0.08] and in the interaction between groups and time [F = 5.50; *p* < 0.001; ηp² = 0.10]. In particular, the FITL group showed an increased perceived effort during the whole training sessions compared to the CTRL group. No significant differences in the groups [F = 1.64; *p* = 0.200; ηp² = 0.3] were found.

### 3.3. Session Enjoyment

The analysis of athletes’ enjoyment (PACES) during training, showed no significant differences in the groups [F(1,47) = 0.07; *p* = 0.797; ηp² < 0.01], time [F(2.2,103.34) = 1.04; *p* = 0.361; ηp² = 0.01] and the interaction between the groups and time [F(2.2,103.34) = 0.614; *p* = 0.558; ηp² < 0.01], as shown in Figure 5.

### 3.4. Fitness Tests

After a period of massed basketball training both groups improved significantly their performances in the fitness tests. 

In the agility measurement (Figure 6A), a significant decrease in the time needed to complete the test was found, revealing an agility improvement in both groups over time [F(1,47) = 11.098; *p* = 0.002; ηp² = 0.19]. No significant differences were found in the groups [F(1,47) = 0.02; *p* = 0.882; ηp² < 0.01] and the interaction between the groups and time [F(1,47) = 0.058; *p* = 0.810; ηp² < 0.01].

In the Yo-Yo IR1 tests (Figure 6B) both groups significantly improved their final scores over time [F(1,47) = 22.47; *p* < 0.001; ηp² = 0.32]. In addition, the Fitlight group scored significantly better in the Yo-Yo IR1 tests than the CTRL group [F(1,47) = 7.23; *p* = 0.010; ηp² = 0.13]. Additionally, a significant difference in the interaction between the groups and time [F(1,47) = 5.09; *p* = 0.029; ηp² = 0.10] was identified, in which the Fitlight group scored higher than CTRL group at T7.

## 4. Discussion

In the current study we analyzed the effect of CMT on cognitive and physical variables in basketball athletes before and after a massed period of training, during which, the perceived effort and enjoyment was also assessed. The main finding of this study was that three weeks of massed basketball training (BT) improved the EFs of athletes independently by the training typology (i.e., basketball drills or Fitlight training). These results are in line with recent articles analyzing the effects of BT on EFs, even if they used different study designs. For instance, Xu and colleagues [41], comparing children (6 to 8 years old) with low and high week BT volume, concluded that the frequency of BT (more than two times per week) was positively associated with an enhancement of EFs. Additionally, Wang and colleagues [18] found that 12 weeks of mini-BT (five days per week) had a positive effect on EF development in children aged 6–12 with autism spectrum disorder compared to a control group (maintenance of normal daily activities). Thus, our findings provide additional information supporting the positive causal correlation between open-skill sports and cognitive progression at the developmental age, according to previous evidence [6,9,42,43].

Although, we did not find any differences between the groups in the EF tasks, we found a significant improvement in both groups over time (from T0 to T8). Additionally, we observed a significant main effect in the interaction between the groups and time in the rate of correct scores in the Forward-Digit Span, an index of working memory processing speed. This result suggests that a Fitlight intervention, in addition to basketball training, could improve working memory [1]. However, because the other measures were not different between the groups, the results need to be interpreted with caution. In general, the results of previous studies have been contradictory regarding transfers between different domains [44] (e.g., Fitlight plus basketball training intervention to EFs).

For instance, Badau and colleagues [27] implemented 3 months of sports training with the Fitlight technology in open-skill sports (basketball, handball and volleyball) in adolescent players and found that, after this training period, the athletes significantly improved their cognitive reaction time, a parameter used to measure cognitive flexibility [27]. Additionally, they found significant improvements in cognitive tests and sports performance after a period of CMT in basketball [1] and open-skill sports players [27]. The different findings could be explained by the different intervention times; in our study the intervention period consisted of 3 weeks of massed Fitlight training, five times per week while Badau and colleagues [27] used a 12-week program for three times a week. Despite a similar total training volume, the distributed cognitive training could represent a necessary condition to achieve significant improvements in EFs during sports practice. For this reason, our 3-week protocol could not induce significant cognitive improvements.

In agreement with our findings, Theofilou and colleagues [45] did not find significant improvements in cognitive outcomes in adolescent soccer players, following 6 months of training intervention with a visual stimuli program. Therefore, the potential benefits of Fitlight training on EFs in open-skill sports needs to be further investigated. In particular, it could be relevant to compare different distributions of cognitive training interventions (massed vs. distributive) in order to understand the optimal volume and frequency for cognitive enhancement.

Moreover, we found that the FITL-trained group experienced higher sRPE—the perceived effort for the whole daily session—among the 3 weeks of intervention. The same results were obtained evaluating the eRPE, which is referred to the intervention alone (perceived effort during the Fitlight session or shooting session alone). This last result leads us to think that the added perceived effort in the whole training sessions is most likely due to the different typology of the intervention. In fact, the FITL group training required higher cognitive effort, which is probably represented in the higher sRPE and eRPE results. For this reason, our results could suggest that the Fitlight training, added to a basketball session, could improve the training effort by enhancing the cognitive demands as shown by both sRPE and eRPE compared to the execution of basketball drills, even if both interventions similarly increase EFs.

From current literature it has been suggested that a higher perceived effort is the strongest indicator of mental fatigue in sport [46]. However, the influence of RPE within the effect of mental fatigue on sport-specific performance remains ambiguous [47]. Although a considerable body of literature supports the idea that mental fatigue induced by cognitive demands can worsen sport performance [47,48,49], including basketball technical and cognitive performance [50,51], recent evidence suggested that under the conditions of pre-induced mental fatigue, athletes can increase the efficiency of their actions and improve their tactical performance, improving action selection and attentive focus [52]. Thus, future research should investigate if mental fatigue induced by Fitlight training leads to an improvement in sport-specific performance.

Enjoyment and motivation [53] could also influence the training progression in sports practice since they are positively associated with improved EFs at the developmental age [17]. In our study we found that PACES did not show any differences between the groups concerning the negative outcomes of the two different training regimes (Fitlight vs. basketball drill sessions). This data, considering the results obtained in the sRPE and eRPE analysis, suggests that the increase in the effort, likely induced by Fitlight training, did not affect the enjoyment of the training sessions.

Regarding the analysis of the fitness tests, a significant improvement in both agility and Yo-Yo IR1 scores after the 3 weeks of massed training was found. However, no significant differences were detected between the groups. The results relative to the Yo-Yo IR1 test, a test assessing metabolic performance, could be explained by the nature of the intervention, which was focused on cognitive and technical skills enhancement rather than metabolic training. Conversely, in the agility tests we expected differences between the groups, in line with previous findings that reported higher values in repeated sprints after the Fitlight training intervention in young soccer players [45]. However, the authors used an intervention period of 6 months for 15 min per day of cognitive training. Thus, if compared to the duration of our protocol, agility could need a longer intervention time to show significant improvements.

This study has some limitations. The subjects included in the study were all male athletes and it would have not been possible to analyze gender differences in CMT and EFs. The time of the massed intervention could have been too short to appreciate any hypothetical differences in EF improvement in response to CMT. However, the 3-week intervention duration was chosen to adapt the intervention with seasonal sports commitments. Moreover, even if validated motor and cognitive tests were used to assess the athletes’ progression, it is difficult to design ecological protocols that could assess and explain how the improvement in EFs affect sports performance during match actions and competitions.

## 5. Conclusions

In conclusion, we found that a 3-week schedule of basketball training improved both cognitive and fitness performance in young basketball players. Nevertheless, CMT by means of the Fitlight training system in addition to this program increased the perceived effort without decreasing enjoyment. For this reason, Fitlight training could be a useful tool to increase cognitive effort without decreasing athletes’ motivation during basketball practice. The use of sensorized lights, such as Fitlights^TM^, could increase training variability management (e. g. light colors, onset stimuli time). Moreover, it allows to set up individualized training programs, enabling coaches to manage small groups of athletes simultaneously, differentiating their training. Hence, these devices can facilitate CMT building and execution, stimulating EFs and improving an athlete’s decision-making strategies.

In the light of this study results, a massed cognitive-motor program of only 3 weeks, seems unable to generate additional improvements in EF development in young basketball athletes. Currently, it is still not clear what the minimum intervention time is to allow significant enhancements in EFs with CMT exercises. From the existing literature 8 to 12 weeks of CMT seems to be able to generate improvements in EFs [27,54] and a distributed schedule seems to be a necessary condition for EF development in open-skill sport. Coaches should consider this when building seasonal training programs.

Many open questions remain regarding EFs and motor training relationships. In future research it could be worth studying which EFs are enhanced by a specific CMT; which could be a motor task to better stimulate EF activation and how long these improvements last. In this context, it will also be crucial to fill the gap between EF development and sports performance, to better understand how having a high level of EFs could impact different sport disciplines.

## Figures and Tables

**Figure 1 ijerph-20-00817-f001:**
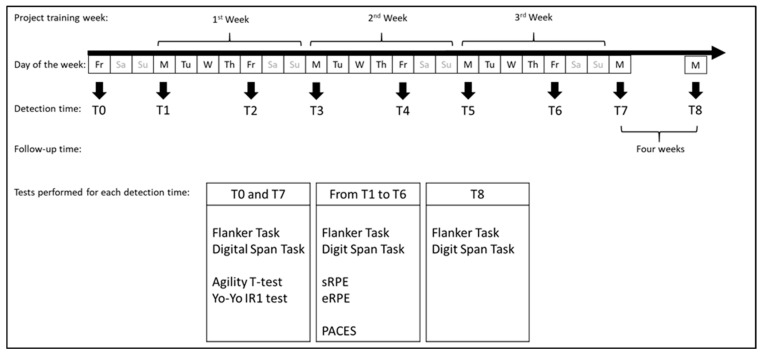
Timeline of the experimental procedures.

**Figure 2 ijerph-20-00817-f002:**
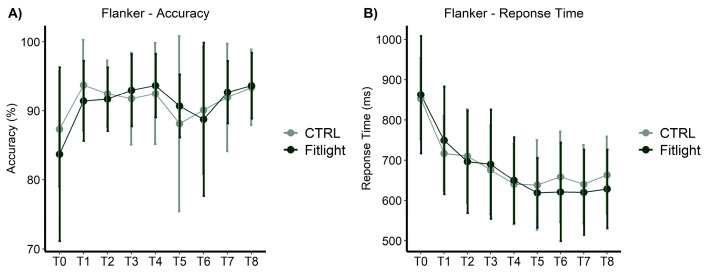
Flanker/Reverse Flanker task measures over time and between the groups in all conditions: (**A**) Accuracy in the Flanker/Reverse Flanker task; (**B**) response time in the Flanker/Reverse Flanker task.

**Figure 3 ijerph-20-00817-f003:**
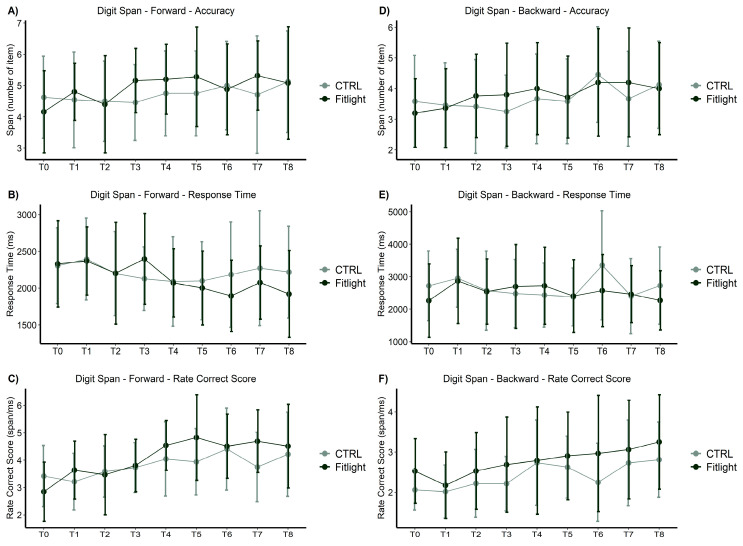
Digit Span measures over time and between groups in all conditions: (**A**) Span measures in Forward-Digit Span; (**B**) response time in Forward-Digit Span; (**C**) rate of correct scores in Forward-Digit Span; (**D**) span measures in Backward-Digit Span; (**E**) response time in Backward-Digit Span; (**F**) rate of correct scores in Backward-Digit Span.

**Figure 4 ijerph-20-00817-f004:**
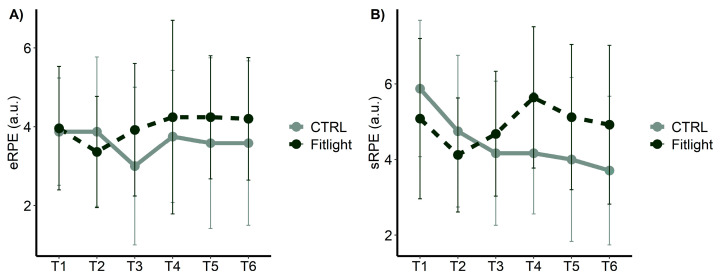
Rate of perceived exertion (RPE) measured over time and between groups in all conditions: (**A**) Exercise ratings of perceived exertion in arbitrary units; (**B**) session ratings of perceived exertion in arbitrary units.

**Figure 5 ijerph-20-00817-f005:**
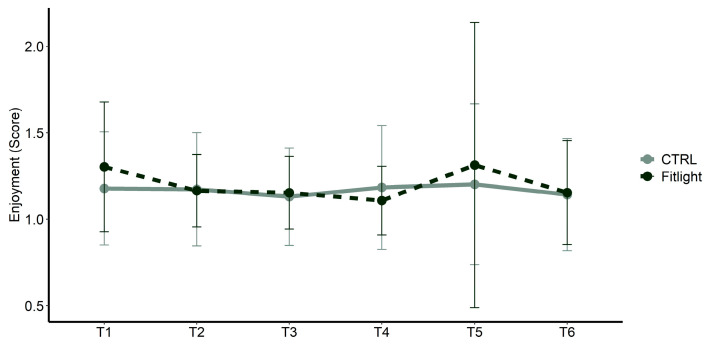
Enjoyment scores measured over time and between groups in all conditions.

**Figure 6 ijerph-20-00817-f006:**
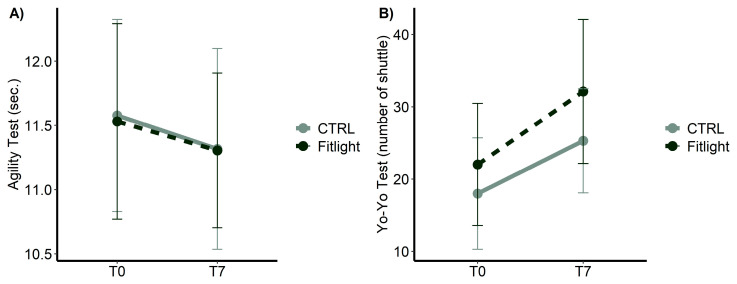
Fitness tests measured over time and between the groups in all conditions. (**A**) Agility test; (**B**) Yo-Yo test.

**Table 1 ijerph-20-00817-t001:** Fitlight training program details.

Week	Type of Exercise	Specific Proposal	Fitlight Role Description
First Week	Footwork Dribbling drills Shooting drills	- Defensive individual footwork drill - Offensive individual footwork drill - 1 vs. 1 dribbling exercise - Basketball dribbling obstacle course - Shooting drill on decision-making	The Fitlight system was used to create a randomized sequence of flashing lights. A sequence of two or three different colors was used up to a maximum of six lights. The association between color and movement changed with each exercise. Each color was associated with a single movement. A specific color did not require any movement in response.
Second Week	Footwork Dribbling drills Shooting drills	- Defensive individual footwork drill - Defensive footwork drill (couple session) - Basketball dribbling obstacle course - Shooting drill on decision-making - Partner shooting drill for spacing	A sequence of three or four different colors was used up to a maximum of eight lights. The association between color and movement changed with each exercise. Each color was associated with a single movement but there were sequences where two colors corresponded to the same movement. Two lights could be switched on at the same time. A specific color did not require any movement in response and this color changed with each exercise.
Third Week	Footwork Dribbling drills Shooting drills	- Offensive footwork drill (couple session) - 1 vs. 1 dribbling exercise - Basketball dribbling obstacle course - Shooting drill on decision-making - Partner shooting drill for spacing	A sequence of four or five different colors was used up to a maximum of 10 lights. The association between color and movement changed with each exercise. Each color was associated with a single movement but there were sequences where two colors corresponded to the same movement. Two lights could be switched on at the same time. Two specific colors did not require any movement in response and these colors changed with each exercise.

## Data Availability

All relevant data are within the manuscript.

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
