# Peer review of "Acute Effects of Fitlight Training on Cognitive-Motor Processes in Young Basketball Players"

_ijerph, 2023, doi:10.3390/ijerph20010817_

Round 1
Reviewer 1 Report
The article entitled "Acute effects of Fitlight training on young basketball players: a cognitive-motor investigation" deals with a very interesting and topical subject whose results can provide findings and considerations regarding possible actions in the training of open skills sports such as, in this case, basketball for the improvement of the variables raised. However, as mentioned, it is necessary to consider some limitations of this study, avoiding generalizing in the conclusions, since the time of application of the research is short.
As contributions and suggestions for improvement, some formal and content aspects are indicated:
Eliminate the final period of the title.
In the key words it is recommended not to repeat terms like those that already appear in the title of the work and in this case to reduce the number of key words: they are excessive. It is recommended between 4 to 6.
Despite the support of scientific literature in Introduction section, it is suggested that the authors increase the ideas developed, mention some aspects that are later developed in the study, as well as order the information according to the proposed content. For example, add something more about Executive Functions and mention the benefits continuously. It is suggested not to include the link to the FITLight program, to put the year of creation (2011) and to move the link to the references.
It is recommended to improve the wording of the objective by going deeper (modifying the infinitive "investigate" for a more specific one (analyze, identify, compare, know.....) and specifying something more in the content of what is being done. Try to provide a little more information.
The paragraph between lines 85-90 seems more appropriate for the conclusions as a proposal for future lines of research, it is suggested to change it to this section.
Regarding Method in section 2.1. it is suggested that it be explained or clarified to which clubs or initial groups the 9 excluded players belong. It leads to confusion. In addition, detail some data such as the level of competition, the criterion for the reorganization of the two groups from the three clubs, ...
In Results section, the title of Figure 6 should be placed below it.
In Discussion section, it would be important to unify the objective with the one stated in the introduction. In this case, emotional aspects are mentioned that did not appear previously. Improve and unify.
In Conclusions section it would be possible to add some more concrete example of concrete orientations for coaches that are extracted from this study, despite the limitations that the research presents. Add the limitations and future lines of work.
The application time is very short, so it should be specified, based on the existing literature, which approach would be the most suitable for the design to be more adequate.
The references section is adequate, specific, and updated.
Reviewer 2 Report
The authors present a paper on "Acute effects of Fitlight training on young basketball players: a cognitive-motor investigation". The intervention study with an active control group (basketball practice) is conducted to study the impact of massed basketball training enriched with Fitlight training elements on Executive functions (EFs) and motor tasks (i.e., agility T-test, Yo-Yo IR1 test). Fitness tests and EFs tasks were conducted with 49 male basketball players. The findings show significant improvements in EFs with no differences between the groups (Fitlight vs. control group). Moreover, an increased RPE was found in the FITL group with no group differences in physical activity enjoyment and fitness tests. As a result, the current study's findings suggest that a 3-weeks of massed basketball training improves EFs and motor performance in young basketball players. The additional Fitlight intervention increases the perceived cognitive effort without decreasing practice enjoyment, even if it seems unable to induce additional improvements in EFs. The study is conducted with high scientific standards, and the manuscript is well-written. However, the reviewer has major concerns about publishing the study in its current form. Some general and detailed comments on the manuscript:
General comments:
1. I think the general theory of the paper has to be strengthened. The study seems not to be based on a specific theory. At this time, the study appears to be just explorative. The authors have to explain the benefits of open-skill sports in detail. Furthermore, they have to explicate how the enrichment of the intervention or training program by using the Fitlight system should improve the EFs of participants more than the common training interventions.
2. In my opinion, the authors should refer to reviews or meta-analyses at some points.
For reviews and meta-analyses, see, for instance:
10.3390/brainsci12081071
10.3389/fpsyg.2019.01707
3. EF measurements: The selection of the tasks should be made clear in the methods section (lines 185-224).
4. The manuscript lacks a limitations section. The authors should clarify which limitations are inherent to the study.
5. Some English language issues:
Line 239: “EFs tests”
More comments in detail:
Abstract:
1. Lines 23-24: Even in the abstract contains relevant information about the study; it lacks a short description of why the authors decided to use this particular study design. It seems to be explorative at the moment.
2. Lines 24-26: The abstract should contain information on the used tasks/tests.
Introduction:
3. Line 48: Please cite a manuscript focussing on the connection between the used tasks and the development of the prefrontal cortex [2].
4. Line 50: There is an important difference between the development and maturation between adolescents and children regarding cognitive functions. Please explain the differences and why you studied this sample of youth basketball players. The particularities of examining EFs of youth game sports athletes could be found for example in the manuscripts of Beavan et al. (2020) or Heilmann et al. (2022):
10.20338/bjmb.v13i2.131
10.1080/1612197X.2021.2025141
5. Line 55: Please report the differences between open and closed-skill sports athletes in general. There are meta-analyses on that topic:
10.3390/brainsci12081071
10.3389/fpsyg.2019.01707
6. Line 78: The examples of cognitive functions are vague. In my opinion, anticipation is not a cognitive function.
7. Lines 86-88: This sentence is confusing.
8. Lines 91-93: Please explain the rationale of the study in a scientific way. The study should test a CMT, not the Fitlights as a product.
9. Lines 94-96: Why did the authors choose inhibition and working memory to examine as EFs? The introduction often lacks in detailed explanation. It is not clear to the reader why the authors did not examine cognitive flexibility if they refer to the definition of EFs by Diamond (2012).
10. Lines 96-98: The hypothesis is not underpinned by the theoretical background in the introduction. Please provide more information. Furthermore, please explain the benefits of the program and its practical implications.
Method:
11. Line 107: Which classification did the authors use for the puberal status?
12. Lines 124-127: Please provide an ethics approval code if possible.
13. Lines 141-144: Please provide a detailed explanation of the training program. I think it is not clear to the reader why this special program should impact EFs.
14. Lines 186-188: Please provide the reliability of the tasks and a more detailed explanation of the tasks. A figure of flanker tasks would be nice.
Discussion:
15. Line 313: Which emotional variable?
16. Lines 317-319: You compare two different study designs. If you want to keep this part, please explain the difference in the study design.
17. Lines 379-380: I think this sentence fits well in the conclusion section.
18. Lines 389-390: Is a citation lacking for this sentence?
Reviewer 3 Report
Reviewers result for the article of ijerph-2095000-peer-review-v1
Major comments
This is the article studying the effects of cognitive-motor training (CMT) for three months period to 49 participants of well-trained basket-ball male players aged 15 years or around this age. The authors described their methods scientifically to details and the results seem analyzed properly. My concerns are the follows:
1. Could these positive results be extrapolated to ordinary playing high school male who have not high performance in basket-ball? The authors did not explain the level of skills of the participants of basket-ball. I suppose that the participants might be relatively higher leveled and the result might not be available to ordinary leveled high school boy. In other words, the availability of the results seems too narrow.
2. It seems unclear what the authors would show to the readers. The impacts of the CMT seems too short because the methods of the last tests were done after not three or six months, but four weeks. This four-weeks interval seems not enough to show long term effects. If the author would show short term effects by their results, they must explain to which area in medical this could be applied.
Minor comments
1. The methods seem difficult to understand. To avoid this, the author might use photographs and/or illustrations to show the methods.
